# Pre-Pandemic Cross-Reactive Immunity against SARS-CoV-2 among Siberian Populations

**DOI:** 10.3390/antib12040082

**Published:** 2023-12-09

**Authors:** Olga N. Shaprova, Daniil V. Shanshin, Evgeniia A. Kolosova, Sophia S. Borisevich, Artem A. Soroka, Iuliia A. Merkuleva, Artem O. Nikitin, Ekaterina A. Volosnikova, Nikita D. Ushkalenko, Anna V. Zaykovskaya, Oleg V. Pyankov, Svetlana A. Elchaninova, Dmitry N. Shcherbakov, Tatiana N. Ilyicheva

**Affiliations:** 1State Research Center of Virology and Biotechnology VECTOR, Rospotrebnadzor, 630559 Koltsovo, Russia; ngelya209@gmail.com (O.N.S.); dan6091154224@gmail.com (D.V.S.); j.a.merkulyeva@gmail.com (I.A.M.); a.nikitin1@g.nsu.ru (A.O.N.); volosnikova_ea@vector.nsc.ru (E.A.V.); ushkalenkonikita@gmail.com (N.D.U.); zaykovskaya_av@vector.nsc.ru (A.V.Z.); pyankov@vector.nsc.ru (O.V.P.); dnshcherbakov@gmail.com (D.N.S.); ilicheva_tn@vector.nsc.ru (T.N.I.); 2Russian-American Anti-Cancer Center, Altai State University, 656049 Barnaul, Russia; 3Laboratory of Chemical Physics, Ufa Institute of Chemistry Ufa Federal Research Center, 450078 Ufa, Russia; monrel@mail.ru; 4Institute of Intelligent Cybernetic Systems, National Research Nuclear University MEPhI (Moscow Engineering Physics Institute), 115409 Moscow, Russia; aasoroka@mephi.ru; 5Department of Biochemistry and Clinical Laboratory Diagnostics, Altai State Medical University, 656038 Barnaul, Russia; saelch@mail.ru

**Keywords:** SARS-CoV-2, antibodies, IgG, coronavirus, pre-existing immunity

## Abstract

In December 2019, a new coronavirus, SARS-CoV-2, was found to in Wuhan, China. Cases of infection were subsequently detected in other countries in a short period of time, resulting in the declaration of the COVID-19 pandemic by the World Health Organization (WHO) on 11 March 2020. Questions about the impact of herd immunity of pre-existing immune reactivity to SARS-CoV-2 on COVID-19 severity, associated with the immunity to seasonal manifestation, are still to be resolved and may be useful for understanding some processes that precede the emergence of a pandemic virus. Perhaps this will contribute to understanding some of the processes that precede the emergence of a pandemic virus. We assessed the specificity and virus-neutralizing capacity of antibodies reacting with the nucleocapsid and spike proteins of SARS-CoV-2 in a set of serum samples collected in October and November 2019, before the first COVID-19 cases were documented in this region. Blood serum samples from 799 residents of several regions of Siberia, Russia, (the Altai Territory, Irkutsk, Kemerovo and Novosibirsk regions, the Republic of Altai, Buryatia, and Khakassia) were analyzed. Sera of non-infected donors were collected within a study of seasonal influenza in the Russian Federation. The sample collection sites were located near the flyways and breeding grounds of wild waterfowl. The performance of enzyme-linked immunosorbent assay (ELISA) for the collected sera included the usage of recombinant SARS-CoV-2 protein antigens: full-length nucleocapsid protein (CoVN), receptor binding domain (RBD) of S-protein and infection fragment of the S protein (S5-6). There were 183 (22.9%) sera reactive to the S5-6, 270 (33.8%) sera corresponding to the full-length N protein and 128 (16.2%) sera simultaneously reactive to both these proteins. Only 5 out of 799 sera had IgG antibodies reactive to the RBD. None of the sera exhibited neutralizing activity against the nCoV/Victoria/1/2020 SARS-CoV-2 strain in Vero E6 cell culture. The data obtained in this study suggest that some of the population of the analyzed regions of Russia had cross-reactive humoral immunity against SARS-CoV-2 before the COVID-19 pandemic started. Moreover, among individuals from relatively isolated regions, there were significantly fewer reliably cross-reactive sera. The possible significance of these data and impact of cross-immunity to SARS-CoV-2 on the prevalence and mortality of COVID-19 needs further assessment.

## 1. Introduction

The isolation of the first human coronaviruses (HCoV-229E and HCoV-OC43) took place almost 50 years ago; two other seasonal coronaviruses (HCoV-NL63 and HCoV-HKU1) were identified only after the SARS-CoV outbreak in 2002–2003 [1,2]. They circulate widely between people and are responsible for a number of respiratory diseases that are usually mild in immunocompetent children and adults. Seropositivity to at least three of the hCoVs is demonstrated to be relevant for more than 90% of the human population [3].

Severe acute respiratory syndrome coronavirus 2 (SARS-CoV-2), which emerged in late 2019 and spread rapidly around the world, is a part of the Betacoronavirus genus and is in closer relation to HKU1 and OC43 than to alphacoronaviruses 229E and NL63 [4]. Observations based on studying electronic medical records showed that pre-existing immune responses to human seasonal coronaviruses did not ensure protection against infection but could soften disease manifestations stemming from SARS-CoV-2 infection [5].

Thus, the level of morbidity and mortality during the first wave of the pandemic reported for Asia, Europe and the United States was different from that in sub-Saharan Africa, where the number of infections and deaths remained relatively low. It was suggested that the previous interaction with other coronaviruses in these populations prior to the COVID-19 pandemic led to cross-protection against SARS-CoV-2 infection. To investigate this hypothesis, plasma samples collected before the COVID-19 pandemic in Tanzania, Zambia and the United States were tested to determine their serological cross-reactivity against the spike and nucleocapsid proteins of SARS-CoV-2 along with a number of different coronaviruses (SARS, MERS, HCoV-OC43, HCoV-HKU-1, HCoV-NL63 and HCoV-229E). African specimens collected before the COVID-19 outbreak were shown to be significantly more serologically reactive against SARS-CoV-2 than American specimens. Also, the samples interacting with SARS-CoV-2 exhibited clear recognition of the spike and nucleocapsid proteins of certain seasonal human coronaviruses [6]. However, the non-neutralizing antibodies that existed before the pandemic and were induced by seasonal hCoV have been shown to provide no protection against SARS-CoV-2, and there has to be a particular focus on evaluating T cell responses against hCoV infections that may play a role in partial protection against SARS-CoV-2 infection [7].

Early reports demonstrated that T cells exhibit significant reactivity in many people who have not faced SARS-CoV-2 previously. At that time, the SARS-CoV-2-specific T cells in such individuals were suggested to be developed from memory T cells derived from seasonal hCoV exposure. Thiel and colleagues [8] reported that T cell reactivity was the highest with respect to the pool of SARS-CoV-2 spike peptides, which had a higher homology with hCoV.

The latest investigations have made it clear that some individuals already had SARS-CoV-2-specific CD4+ and CD8+ T cells before the COVID-19 outbreak took place [8,9,10,11,12]. Also, the pre-existing cellular immunity was considered to serve as one of the essential factors of protection against pandemic viruses that have common epitopes for which non-neutralizing antibodies are produced.

Therefore, antibodies against seasonal coronaviruses that possess no neutralizing ability to SARS-CoV-2 presumably play no role in protecting a person against COVID-19, but the presence of such antibodies in the blood serum proves the fact of recent infection with seasonal viruses and, therefore, the presence of T cell-mediated immunity ensuring partial protection against the severe course of COVID-19.

The aim of this study was to test human sera from the Siberia region of Russia collected prior to the beginning of the COVID-19 pandemic. We evaluated the interaction of antibodies with spike and nucleocapsid antigens, which allowed us to correlate this IgG response to SARS-CoV-2 neutralization.

## 2. Materials and Methods

### 2.1. Serum Samples

Seven hundred ninety-nine blood serum samples were taken in October and November 2019 by staff of the Centers for Hygiene and Epidemiology in the following regions of the Russian Federation: the Altai Territory, Irkutsk, Kemerovo and Novosibirsk Regions, and the Republics of Altai, Buryatia and Khakassia. Sera were collected as part of a study focusing on seasonal influenza in the Russian Federation. The sample collection sites were located close to waterfowl breeding and habitat areas. The collection of non-infected donors’ sera and its transportation to the State Research Center of Virology and Biotechnology VECTOR, Rospotrebnadzor (Koltsovo, Russia) were carried out according to a procedure described previously [13]. In addition, we used 29 serum samples collected between February and October 2021 that belonged to patients who had recovered from COVID-19 in the Altai Territory, for whom the diagnosis was verified by PCR assay.

### 2.2. Construction of Expression Plasmids pET21-S5-6, pET21-CoVN and pVEAL2-RBD

The recombinant proteins S5-6 (protein fragment S, including amino acids F718 through G908), CoVN (protein corresponding to the full-length viral protein N) and RBD (protein fragment S corresponding to the receptor-binding domain and including amino acids V308 through N542) were used in this study. Nucleotide sequences encoding SARS-CoV-2 S5-6, CoVN and RBD proteins were constructed on the base of the Wuhan-Hu-1 strain, GenBank: MN908947. The codon composition of nucleotide sequences for expression in *Escherichia coli* and mammalian cells underwent optimization in the GeneOptimizer tool (https://www.thermofisher.com/ru/en/home/life-science/cloning/gene-synthesis/geneart-gene-synthesis/geneoptimizer.html (accessed on 5 September 2020)). The final nucleotide sequences were synthesized as part of the pGH vector plasmid (OOO DNK-Sintez, Moscow, Russia).

The plasmid vector pET21-S5-6 was derived from the pET21 vector. The nucleotide sequence encoding S5-6 was amplified from the pGH-S5-6 template with S5-F (5’-aaaaaaGGATCCTTCACCATCAGCGTGACCACAG-3’) and S6-R (5’-aaaaaaCTCGAGGCCGTTGAACCGGTAGGCC-3’) serving as primers. The product of PCR was subsequently cloned into the expression vector pET21 at the BamHI and Sfr274I restriction sites in frame with 6 × His.

For amplification of the DNA fragment encoding CoVN from the pGH-CoVN template, the primers CoVN-F (5′-aaaaaaggatcctctgataatggaccccaaaatcagc-3′) and CoVN-R (5′-aaaaaagcggccgcggcctgagttgagtcagca-3′) were used. The amplified fragment was then cloned into the expression vector pET21 at the BamHI and NotI restriction sites in frame with 6 × His.

In order to obtain the RBD fragment (region 308 V–542 N) of the SARS-CoV-2 spike protein (Wuhan-Hu-1, GenBank: MN908947), it was decided to use the transgenic cell culture that contained the relevant gene. Integration was performed using the pVEAL-RBD vector [14]. Optimization of the DNA codon composition for expression in mammalian cells was performed in the GeneOptimizer tool (https://www.thermofisher.com/ru/en/home/life-science/cloning/gene-synthesis/geneart-gene-synthesis/geneoptimizer.html (accessed on 5 September 2020)). The pGH vector plasmid containing the resulting nucleotide sequence was synthesized (OOO DNK-Sintez, Russia), and the gene was cloned into the pVEAL2 vector. The N-terminal region of RBD was presented by a sequence responsible for tissue plasminogene activator, Tpa (MDAMKRGLCCVLLLCGAVFVSA), while the C-terminus included the sequence 6×His.

### 2.3. Preparation and Purification of S5-6, CoVN and RBD Proteins

Recombinant plasmids pET21-S5-6 and pET21-CoVN served to transform *E. coli* strain BL21(DE3) cells [15]. The *E. coli* cells were seeded on nutrient medium containing 100 μg/mL ampicillin. After transformation, the colony of each producer was transferred to 5 mL of a nutrient medium containing 100 μg/mL of ampicillin and cultured in an Innova40 thermostatically controlled rotary shaker (NB, Edison, NJ, USA) at 160 rpm overnight at 37 °C (an overnight culture). The next day, the overnight culture in a volume of 1.5 mL was placed into 150 mL of fresh nutrient medium, and each producer was cultivated in a thermostatically controlled shaker at 180 rpm at 37 °C until OD_600_ = 0.6–0.8. IPTG (Isopropyl β-d-1-thiogalactopyranoside) was then added to a final concentration of 1 mM as an inducer and cultivated for another 16 h.

The biomass was obtained by centrifugation on Thermo Scientific SL8 (Thermo Fisher Scientific, Waltham, MA, USA) at 6000× *g* for 10 min at 4 °C and then resuspended in lysis buffer (20 mM imidazole, 30 mM NaH_2_PO_4_, 500 mM NaCl, 8 M urea, 0.1% Triton X-100, pH 7.4) and disintegrated on Alena series ultrasonic apparatus UZTA-0.15/22-O (LLC SPE Biomer, Novosibirsk, Russia for 10 cycles lasting 90 s each, with a break of 3 min on ice. The cell lysate was also exposed to 25 min long centrifugation at 15,000× *g* and 4 °C to remove debris. Purification of proteins S5-6 and CoVN was performed by affinity chromatography on a Ni-IMAC column (GE Helthcare, Chicago, USA). Cell lysate was loaded onto the column at a 1.5 mL/min flow rate. The column was washed with a fivefold volume of washing buffer (40 mM imidazole, 30 mM NaH_2_PO_4_, 500 mM NaCl, 8 M urea, pH 7.4) at a 2 mL/min flow rate to remove unbound proteins. Target proteins were eluted with a threefold volume of elution buffer (500 mM imidazole, 30 mM NaH_2_PO_4_, 500 mM NaCl, 8 M urea, pH 7.4) at a flow rate of 1 mL/min. The purity and homogeneity of the obtained proteins were analyzed by 15% SDS-PAGE using the Gel-Pro Analyzer (Media Cybernetics, Rockville, MD, USA), Ver. 3.1 software. Protein concentrations were measured by the Lowry method.

Chinese hamster ovary cell line (CHO-K1) (accession number 30, obtained from the collection of cell cultures SRC VB Vector, Koltsovo, Russia) were exposed to transfection using the pVEAL2-RBD and helper plasmid pCMV (CAT) T7-SB100 that contained SB100 transposase. The transfection was carried out with the use of Lipofectamine 3000 (Invitrogen, Carlsbad, CA, USA). Transfected cells underwent the process of selection with puromycin (10 µg/mL). After that, high-producing clones were extracted by dilution cloning and cultured in roller bottles at 37 °C on DMEM/F-12 (1:1) medium supplemented with 2% FBS and 50 µg/mL gentamicin.

The recombinant RBD was isolated from the culture medium of CHO-K1 cells. The culture medium was centrifuged to remove cell debris and filtered using filtration systems (0.22 μm). The first stage of recombinant RBD purification included metal chelate chromatography on a Ni-NTA column (Qiagen, Germany) equilibrated with 30 mM NaH_2_PO_4_, 20 mM imidazole, 0.5 M NaCl, pH 7.4. The target protein fraction was eluted with a gradient of 20 to 0.5 M imidazole in 30 mM NaH_2_PO_4_, 0.5 M NaCl, pH 7.4.

The next stage started with the purification performed by ion-exchange chromatography on serially jointed columns packed with cation exchange (SP sepharose) and anion exchange (Q sepharose) sorbents equilibrated with 50 mM NaHCO_3_, pH 7.6. The protein was loaded into the columns, and they were washed with NaHCO_3_ (50 mM), pH 7.6. Target protein fractions with an OD of 0.25 were exposed to 15% SDS-PAGE. Impurity proteins on the Q and SP sepharose sorbents were eluted with a linear NaCl concentration gradient from 0 to 1 M in 50 mM NaHCO_3_, pH 7.6.

The resulting RBD specimen was filtrated (0.22 μm). The purity and homogeneity of the obtained protein were analyzed by electrophoresis under denaturing conditions in 15% PAGE using the Gel-Pro Analyzer, Ver. 3.1 software (Media Cybernetics, Rockville, MD, USA). Protein quantification was carried out using the Lowry method.

As a result, recombinant proteins were obtained with a purity of at least 95% and concentrations 1 mg/mL (RBD), 2 mg/mL (S5-6) and 2.5 mg/mL (CoVN).

### 2.4. ELISA Using RBD, S5-6 and CoVN Proteins

Recombinant RBD, S5-6 and CoVN proteins were immobilized on strong adsorption flat-bottomed plates (Nunc, Rochester, NY, USA) at the amount of 200 ng/well in a volume of 100 µL in sodium phosphate buffer (PBS) (Greiner Bio-One, Kremsmunster, Austria) and kept overnight at 4 °C. The wells were washed three times on a PW40 microplate washer (Bio-Rad Laboratories, Hercules, CA, USA) with 300 µL of wash solution containing 0.5% polysorbate 20 in PBS, and 150 µL/well of blocking solution (1% casein in wash solution) was added. The incubation of the plates was then performed in a Binder BD 53 dry-air thermostat (Binder GmbH, Tuttlingen, Germany) for 1 h at 37 °C. Having been washed three times, sera (100 μL) diluted 1:100 with a blocking buffer solution were applied and incubated for 1 h at 37 °C. After washing, HRP-conjugated goat anti-human IgG antibodies (Sigma-Aldrich, St. Louis, MO, USA) at a dilution of 1:20,000 were applied (100 µL/well) and incubated for 1 h at 37 °C. After that, the samples were washed three times, and 100 μL of TMB (3,3’,5,5’-Tetramethylbenzidine) substrate solution (stabilized solution of 3,3’,5,5’-tetramethylbenzidine hydrochloride in a substrate buffer with 0.02% hydrogen peroxide, at a ratio of 1:10, respectively) was applied. After 20 min, 50 μL of 1 N hydrochloric acid was utilized to halt the reaction. The signal was recorded at a λ = 450 nm using a Thermo Scientific Varioscan LUX microplate reader.

Blood sera from individuals diagnosed with COVID-19 verified by PCR were used as a positive control. ELISA was performed twice with each serum sample, where recombinant proteins S5-6, CoVN or RBD were used as an antigen. The mean OD value was calculated for each serum. The median and mean OD values of the test sera were calculated.

### 2.5. Analysis of Virus-Neutralizing Activity of Immune Sera

The neutralizing properties of blood serum antibodies were tested in vitro by measuring the ability of the sera to inhibit the virus cytopathic effect (CPE) on Vero E6 cell culture.

The cell culture was grown in 96-well culture plates in MEM (Minimum Essential Medium) growth medium supplemented with 10% FBS until 100% confluence.

Serial twofold dilutions of the analyzed sera in MEM (1:10–1:5120) were mixed equally with a solution of the SARS-CoV-2 strain nCoV/Victoria/1/2020 (GenBank: MT007544.1) (state collection of the causative agents of viral infections and rickettsiosis of the SRC VB VECTOR, Koltsovo, Russia) in MEM (100 CPE50). The mixture was incubated for 1 h at room temperature and then transferred to the monolayer of Vero E6 cells. The plates were kept for 4 days at 37 °C, 5% CO_2_. Cells were dyed by adding 150 μL of 0.2% gentian violet per well. After 30 min, the liquid was emptied from the wells, which were then washed with water. The results were evaluated visually, taking into account any specific damage to the cell culture in the well as a CPE.

Serum titer was considered to be the dilution at which protection was observed in 50% of wells containing the cell culture from the CPE of the virus. The Reed–Muench formula was applied to calculate the titer of neutralizing antibodies [16].

### 2.6. Statistical Analysis

Optical density values characterizing the interaction of blood serum antibodies with recombinant nucleocapsid N-protein (CoVN) and a part of S-protein (S5-6) of SARS-CoV-2 were used to calculate their normal distributions using the EM (expectation–maximization) algorithm [17,18]. Implementation of the EM algorithm was carried out with the help of the scikit-learn machine learning library based the Python programming language. Statistical analysis by region was performed on a sample of values obtained by ELISA with blood sera from healthy people collected before the onset of COVID-19. The optical density values obtained for the blood sera of people who had SARS-CoV-2 infection were utilized to find the limit of optical density values with significant cross-reactivity.

## 3. Results

A total of 799 blood serum samples collected in October and November 2019 in the Siberian region of the Russian Federation (the Altai Territory, Irkutsk, Kemerovo and Novosibirsk Regions, and the Republics of Altai, Buryatia and Khakassia) were analyzed by enzyme-linked immunosorbent assay (ELISA). Twenty-nine blood sera from individuals diagnosed with COVID-19 verified by RT-PCR were used as a positive control. ELISA was performed twice for each blood serum sample, where the recombinant protein S5-6 (including amino acids F718 through G908 of spike protein S), nucleocapsid N protein (CoVN) and spike receptor-binding domain (RBD) of SARS-CoV-2 were used as antigens (Figure 1). These proteins were chosen for this study because the S and N proteins are the ones involved into the humoral immune response [19,20,21]. Protein S5-6 was included in this study due to the fact that, according to a number of publications, this region (F718 through G908) comprises the S2′ proteolysis site and the fusion peptide, which are important for viral fitness and are the targets of the humoral immune response. This region was also shown to contain sites retaining conservatism with human coronaviruses [22,23].

According to the results of this study, quite a few significantly positive cross-reactive sera interacting with RBD were identified (0.6%). In contrast, a high percentage were reactive against the other two antigens.

The mean optical density (OD) values of the COVID-19 positive blood sera were 0.389 and 0.703 for S5-6 and CoVN, respectively. For the tested sera, the median and mean OD values were 0.213 and 0.241 for S5-6 and 0.232 and 0.264 for CoVN. Despite the low mean values, some of the samples in the test sera set had OD values equal to the COVID-19-positive samples. In contrast, only five (0.6%) sera had an OD value above the threshold for RBD (Figure 2C); therefore, they were not taken for further analysis. In order to identify true positive samples from the test sera, a statistical approach was used. In the set of test sera, the distribution of OD values is uneven; the resulting histograms have an unclear second peak for both CoVN and S5-6 antigens (Figure 2). Statistical analysis of the cumulative optical density data obtained for the analyzed sera shows that the OD values have the highest probability in the range from 0.15 to 0.275 for CoVN and in the range from 0.1 to 0.225 for S5-6. Sera giving such a signal represent the majority, and in general, statistical groups are presumably negative (Figure 2A,B). At the same time, a fuzzy second peak in the range from 0.425 to 0.475 units for CoVN and in the range from 0.375 to 0.450 units for S5-6 may indicate the presence of a group of people whose sera actually have cross-reactivity to SARS-CoV-2 antigens.

This distribution is not uniform across regions; an assessment of the probability density distribution shows a different picture. Thus, for the sample from the Novosibirsk region, the number of potentially cross-reactive sera that contain antibodies reactive to CoVN and S5-6 is greater (Figure 3A,B, brown lines) than for sera obtained from the Republic of Khakassia (Figure 3A,B, blue lines). Optical density values for blood sera from the Republic of Khakassia remain within the noise range for both antigens.

To clarify the result obtained, it was decided to use the EM algorithm for the Gaussian Mixture Model. The indicator of reliable cross-reactivity was assessed by analyzing the OD values of sera of healthy patients before the pandemic and those who had recovered using ELISA for CoVN and S5-6 proteins. Using the EM algorithm, a group of people was separated whose sera were not reliably reactive against CoVN and S5-6 (shown in blue dots in Figure 4). Optical density values of 0.3 (CoVN) and 0.348 (S5-6) were chosen as indicators. The orange dots in Figure 4 correspond to the group of people with probable cross-immunity. To verify these results, red dots corresponding to the blood sera of people who recovered from SARS-CoV-2 were additionally added to Figure 4.

Thus, analysis of statistical data made it possible to determine the sign of cross-immunity. Orange dots falling in squares II–IV correspond to a group of people whose blood sera might exhibit cross-reactivity against recombinant SARS-CoV-2 proteins.

The analysis showed that the number of cross-reactive sera ranged from 22.9% (S5-6) to 33.8% (CoVN) depending on the antigen used. Moreover, the number of sera with cross-reactivity against both antigens is 16% (Table 1).

We also performed a classical virus neutralization assay on all 799 sera. The neutralization titer was determined in the reaction of in vitro inhibition of the cytopathic effect (CPE) of the virus on cell culture. The SARS-CoV-2 coronavirus strain nCoV/Victoria/1/2020 was used for this. None of these sera had neutralizing activity. For all immune sera, a virus neutralization assay was also performed. Virus-neutralizing activity was detected in all samples; titer values ranged from 1/20 to 1/160.

## 4. Discussion

Since the beginning of the COVID-19 pandemic, questions have arisen regarding pre-existing immunity to SARS-CoV-2. Several laboratories have identified antibodies binding to proteins of the pandemic virus as a result of examination of human sera collected long before the emergence of a new pathogen. Thus, Anderson et al. [7] found that slightly more than 4% of serum samples from in the United States in 2017 possessed IgG antibodies that reacted to the SARS-CoV-2 full-length spike protein in ELISA, and at least 16% of serum samples included antibodies that bound to the nucleocapsid protein, but only about 0.9% of the samples were reactive to the RBD domain of the S protein. In addition, individuals who had pre-pandemic antibodies binding to SARS-CoV-2 had increased levels of antibodies to seasonal betacoronaviruses, suggesting that antibodies capable of interacting with the pandemic virus were produced in response to infection with seasonal betacoronaviruses. However, most pre-pandemic cross-reactive antibodies demonstrated no neutralization of the virus and were not able to protect against the SARS-CoV-2 infection [24].

Nonetheless, the functional contribution to the immunopathogenesis of antibodies specific to seasonal coronaviruses remains poorly understood. Although monoclonal antibodies targeting common epitopes in the S protein domain of SARS-CoV-2 and seasonal betacoronaviruses have been identified [25,26], little is known about how antibody reactivity to pandemic and seasonal viruses affects the immune response to the pandemic virus. There is evidence that individuals who had coronavirus infection shortly before the pandemic with OC43 and HKU1 antibody levels but did not possess SARS-CoV-2 antibody levels exhibited a reduced symptom duration in comparison with those who were infected with SARS-CoV-2 for the first time. The cross-protection provided by common betacoronaviruses is likely not mediated by rare antibodies that cross-react to SARS-CoV-2 proteins. Instead, this protection might be mediated by cellular immune responses, which can target epitopes that are conserved among common betacoronaviruses and SARS-CoV-2. It is possible that T cells stimulated from recent betacoronavirus infections are involved virus elimination and reducing symptom duration following SARS-CoV-2 infections.

Thus, in a study by Grifoni et al. [9], T cell reactivity to SARS-CoV-2 proteins was shown to be present in 50% of donated blood samples obtained in the United States before the pathogen had appeared in the human population. A blood donation study in the Netherlands revealed CD4+ T cell reactivity against the SARS-CoV-2 S protein in one out of ten healthy donors [27]. A study conducted in Germany [8] reports 34% positive T cell reaction to S-protein in the donors who were seronegative to SARS-CoV-2. Eventually, a human study in Singapore [10] showed that in 50% of patients with no history of SARS or COVID-19 and no exposure to SARS or COVID-19, T cells responded to the nsp7 or nsp13 nucleocapsid protein. Meckiff et al. [28] worked with samples from the UK and also observed T cell reactivity in individuals who had not had SARS-CoV-2. Therefore, a number of studies have revealed the presence of pre-existing T cells recognizing SARS-CoV-2 in a significant proportion of people from different geographic regions [12]. This potential preexisting cross-reactive T cell immunity to SARS-CoV-2 has broad implications because it could explain aspects of differential COVID-19 clinical outcomes, influence epidemiological models of herd immunity, or affect the performance of COVID-19 candidate vaccines. However, the impact of cellular immunity elicited by prior coronavirus infections on SARS-CoV-2 infections is still poorly understood.

Since circulation of all the seasonal respiratory viruses, except for rhinoviruses, decreased abruptly during the pandemic [29], the level of T cell immunity formed in response to previous infections with betacoronaviruses apparently decreased in people over time. Along with the appearance of more easily transmitted variants, this may also be an explanation for the very few child cases of COVID-19 during the first wave of the pandemic and relatively higher amount of (35–37%) asymptomatic cases of SARS-CoV-2 infection [30].

Analysis of samples collected in Russia showed that coronaviruses circulated among wild and domestic birds. Only gammacoronaviruses were detected among poultry, whereas both gammacoronaviruses and deltacoronaviruses were detected among wild birds. The percentage of coronavirus detection among wild birds was 14.2% and among poultry was 7.3% [31]. In addition, the percentage of virus carriers among wild ducks in Sweden reached 18.7% [32] and in Australia 15.3% [33]. Wild migratory birds appear to play a significant role in the spread of coronaviruses. In this regard, we examined the sera of people living in regions of Russia located on the migration routes (flyways) of wild birds.

The results obtained in this study are in good agreement with the findings reported by other researchers. Thus, among the 799 blood serum samples collected in different regions of Western Siberia in autumn 2019, before SARS-CoV-2 emerged in Russia, 22.9% of the samples were associated with a fragment of the S-protein S5-6 of the pandemic virus in ELISA, 33.8% samples were positive for the full-length CoVN nucleocapsid protein, 16% of the samples were positive for both S5-6 and CoVN proteins, and only 0.6% of the samples were bound to RBD in ELISA. This may be due to the fact that RBD is less conserved than CoVN and S5-6.

Subsequently, none of the examined sera neutralized SARS-CoV-2 in the VERO E6 cell culture.

Interestingly, the proportion of cross-reactive sera varies across regions. Thus, for a sample from the Novosibirsk region, the number of sera reactive to CoVN and S5-6 increases. The maximum peak appears at an OD value of 0.4 in the distributions for both antigens. In the case of the S5-6 assay, the graph (Figure 3B) shows a projection of 0.6 units, indicating that more individuals have antibodies that interact with this region of the spike protein. The Novosibirsk region is a fairly large region of the Russian Federation, on the territory of which the large international airport Tolmachevo is located, as well as the West Siberian Railway. A large number of people and population migration lead to the spread of various viral infections, including coronaviruses and strains of the pathogen SARS-CoV-2. For this reason, the observed distribution is logical (significant). A completely opposite position is observed in fairly isolated regions, such as the Republic of Khakassia and the Kemerovo region (Figure 3A,B, blue and green lines, respectively). The optical density values of blood sera from these regions remain within the noise range for both antigens. The proportion of cross-reactive sera for CoVN in the population of the Altai Republic also appears to be minimal, with a low peak characteristic of values up to OD 0.3 (Figure 3A, purple line). When analyzing data on binding to the S5-6 protein, the peak shifts to 0.6 (Figure 3B). It should be noted here that the N protein is more conserved. The presence of antibodies reactive to this protein may indicate a previous seasonal coronavirus infection, in contrast to antibodies to the less conserved protein S5-6.

Perhaps the previous immunity to SARS-CoV-2 in some people is explained not only by the family ties of the pandemic virus with seasonal coronaviruses, but also by the fairly frequent contact of people with intermediate variants of the virus that appear in the natural reservoirs of animals. This could also explain the different distribution of antibodies to S5-6 across the regions of Siberia.

Our study had a number of limitations. The participants in our study were residents of the administrative territories of Siberia in Russia. This limits the possibility of extrapolating the results to populations in other territories, which may differ, in particular, in the epidemiology of coronavirus infections that preceded the COVID-19 pandemic.

Another limitation of our study is the use of a limited number of antigens. We used the full-length nucleocapsid protein and fragments of the spike protein, covering only part of the epitopes of the latter. It can be assumed that measuring antibody levels to a wider range of SARS-CoV-2 antigens will allow a more accurate characterization of pre-existing humoral immunity to COVID-19.

Thus, our study showed a high percentage of Siberian residents with cross-reactive antibodies to conservative coronavirus antigens. The number of positive sera directly correlated with the degree of protein conservation. Regions of high migration and population density had more antigen-positive individuals, as expected. We assume that the individual immunological background should be taken into consideration of important factors in predicting the response of the immune system to a pathogen, whether it is an infection or vaccination.

Further aims of our work will be to study the sera of the collected samples using a larger number of SARS-CoV-2 antigens. For this, both the full-length spike protein and its fragments will be used, and possibly also other viral proteins. In addition, we plan to include seasonal coronavirus antigens in our studies. Further research in this direction may help identify factors of interindividual variability in the level and duration of persistence of antibodies specific to SARS-CoV-2 antigens and also, possibly, find explanations for some epidemiological features of COVID-19, in particular, the lower incidence of COVID-19 in the pediatric population.

## Figures and Tables

**Figure 1 antibodies-12-00082-f001:**
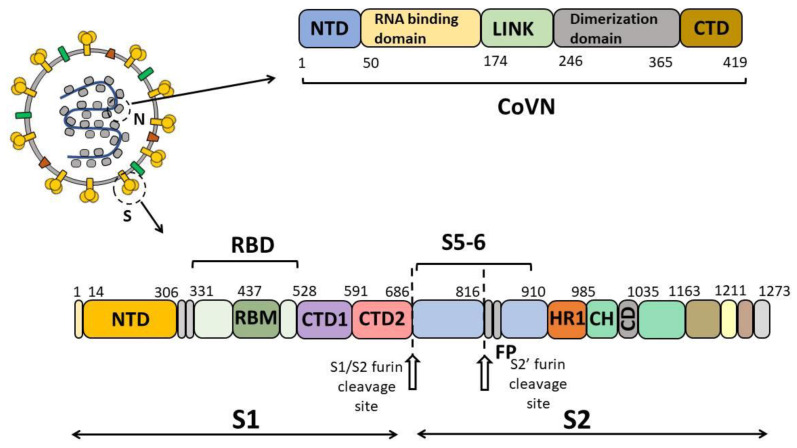
Schematic representation of SARS-CoV-2 antigens (CoVN, RBD, and S5-6) used in this study.

**Figure 2 antibodies-12-00082-f002:**
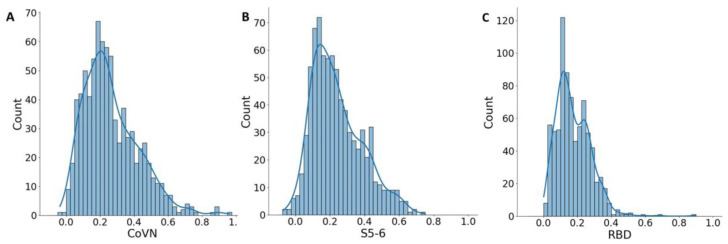
Histogram of the optical density values of the interaction of healthy blood sera in relation to two antigens (CoVN) (**A**), S5-6 (**B**) and RBD (**C**). Range (column) width 0.025.

**Figure 3 antibodies-12-00082-f003:**
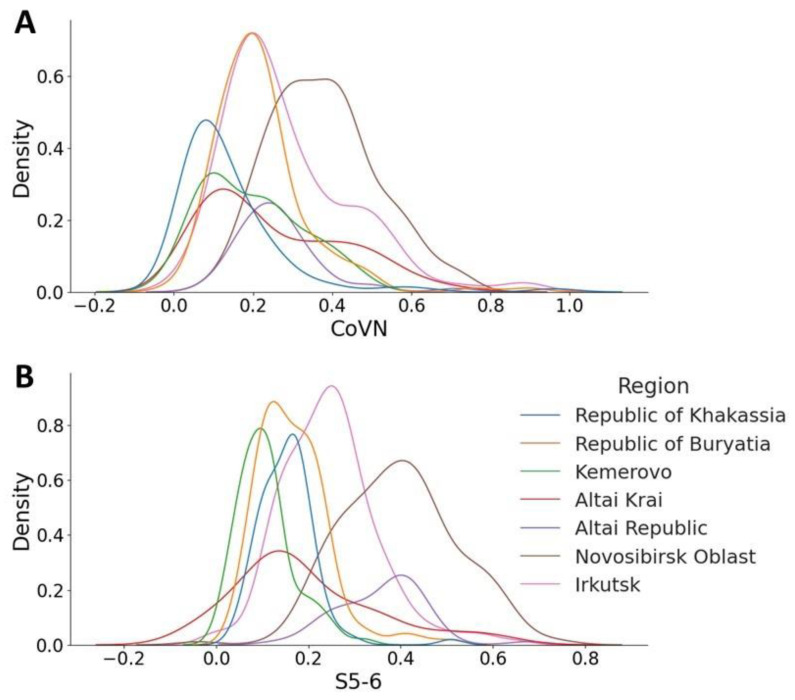
Probability density distribution of the optical density of interaction of blood sera in relation to two antigens CoVN (**A**) and S5-6 (**B**) SARS-CoV-2.

**Figure 4 antibodies-12-00082-f004:**
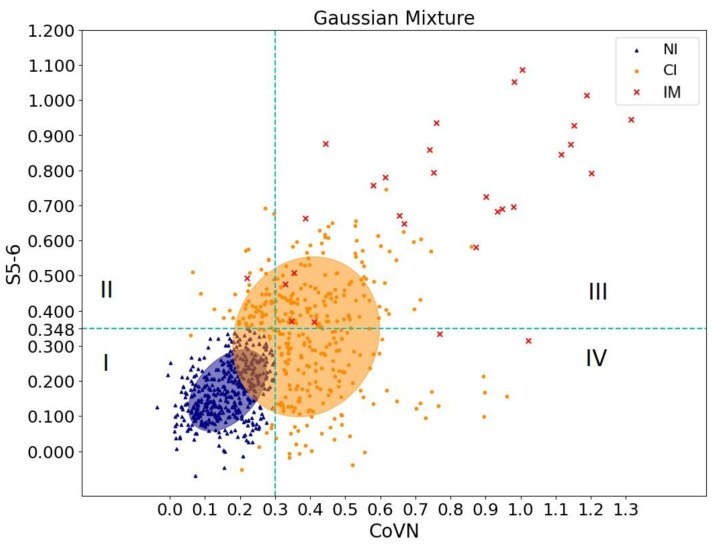
Scatter plot of three groups: NI (not immune) corresponds to the optical density values of blood serum in a group of people without cross-immunity. CI (cross-immune) corresponds to those with probable cross-immunity. IM (immune) corresponds to the group of recovered patients from SARS-CoV-2. The ellipses reflect the standard deviation around the mathematical expectation. The blue dashed lines are constructed at optical density values of 0.3 (CoVN) and 0.348 (S5-6), separating non-cross-immune people (I) from people with cross-immunity positive for S5-6 (II), CoVN (IV) or both of these antigens (III).

**Table 1 antibodies-12-00082-t001:** Number of blood sera with cross-reactivity against SARS-CoV-2 antigens.

Sera	CoVN pos	S5-6 pos	CoVN pos, S5-6 pos	CoVN neg, S5-6 neg
NI: not immune, *n* (%)	0	0	0	346 (43.3%)
CI: cross-immune, *n* (%)	270 (33.8%)	183 (22.9%)	128 (16.0%)	0
IM: immune, *n*	29	29	29	0

## Data Availability

Data is contained within the article.

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
