# Peer review of "Pre-Pandemic Cross-Reactive Immunity against SARS-CoV-2 among Siberian Populations"

_2073-4468, 2023, doi:10.3390/antib12040082_

Round 1

Reviewer 1 Report

Comments and Suggestions for Authors

The authors investigated the antibody response of sera obtained from people resident in Siberia before the COVID-19 pandemic by in-house produced ELISA coated with SARS-CoV- 2 CoVN.

The authors uncovered a weak cross-reactivity in the sera of people from the Republic of Khakassia.

The abstract and the manuscript do not overlap, the result section does not coincide with the material and method section.

Introduction: The author could explain the reason for the collection of sera from the sites located near the flyways and breeding grounds of wild waterfowl.

Line 113 “we used 18 serum samples collected from the recovered patients with COVID-19 in the Altai Territory, for whom the diagnosis was verified by PCR assay.” Please add the collection period date.

Lines 113 and 255: there is a discordance between the number of sera from individuals diagnosed with COVID-19 verified by RT-PCR were used as a positive control. 18 or Twenty-nine samples?

Table 1: Should be revised. Could be added one column for CoVNneg S5-6 neg cases. Please see the attachment. The authors described NI (not immune) corresponding without cross-immunity, for that reason, they should be negatives for these antigens (Figure 4).

The authors should present the results of the antibody response to RBD by ELISA and the virus-neutralizing activity.

Line 410: It needs correction for the typo “We assumet hat the immune”.

Reviewer 2 Report

Comments and Suggestions for Authors

line 24: express better, ...."prevalence "????: I suppose you mean"questions about inpact on covid severity of heard immunity of pre existing immune reactivity to sars cov-2 , associated with the immunity to seasonal manifestetion, are still be resolved and may be usefull to understending  some processes that precede the emergence of a pandemic virus

line 138: primers used?

line 165: elimine RBD, I suppose that it is an error

line 172: the fractions of targets proteins (S5-6 and CoVN) were analyzed by electrophoresis and in which condition for GEL-Pro analyzer? (missing). Indicate please di purity and the concentration of proteins obtained. If you have protein electrophoresis images I suggest to insert them to give more value to the work and lighten the work which only shows statistical data. the same obsarvation are true for the RBD protein (indicate purity and concentarion obtained and show gel eletrhophoresis if it is  possible )

line 174: indicate please negative ed positive controlls used in transefection

line 194 : repetition (error) or a second electrophoresis was made? It is not clear

line 197: analized by GEl Pro on gel eletrophoresis?

Line 186 to 198: decribe better which fraction were analized and how many tmes by gel elettrophoresis and Gel Pro Analyzer. It is not clear or I dont' understend

line 2.5: If you have an immagine of citopatic effect of  virus on cell culture (positive controls )and an immagine  with a virus and  serum dilution titer able to  protect (50%) the cell culture from the CPE of the virus,  introduce to give more value to the wark (optional)

line 350: It is not clear : explain better and verify the affermation " individualas who had had coronavirus infection shortly before the pandemic got sick more easly in the beginning of covid pandemic (????).

Perhaps you mean " there are evidence that individuals who had coronavirus infection shortly before the pandemic with OC43 and HKU1 antibody levels but not SARS-CoV-2 antibody levels was  associated with reduced symptom duration in individuals who are infected with SARS-CoV-2 for the first time.  the cross-protection afforded by common βCoVs is likely not mediated by rare antibodies that cross-react to SARS-CoV-2 proteins. Instead, this protection might be mediated by cellular immune responses, which can target epitopes that are conserved among common βCoVs and SARS-CoV-2 . It is possible that T cells stimulated from recent βCoV infections are involved with clearing virus and reducing symptom duration following SARS-CoV-2 infections. Recent βCoV infections might also  stimulate rare B cells that are quickly recalled following SARS-CoV-2 exposures or that mucosal antibodies elicited by prior CCV infections might be  involved in protection.

line 365 add : This potential preexisting crossreactive T cell immunity to SARS-CoV-2,  has broad implications because it could explain aspects of differential COVID-19 clinical outcomes, influence epidemiological models of herd immunity, or affect the performance of COVID-19 candidate vaccine ; however, the impact of cellular immunity elicited by prior CCV infections on SARS-CoV-2 infections is still  poorly understood

Finally: explain the future aims and eventally limitation of the study

Comments on the Quality of English Language

Minor editing of english language

Reviewer 3 Report

Comments and Suggestions for Authors

Dear Authors: Congratulations on this excellent work. This is an excellent piece of research and its results and conclusions have been explained in a clear and concise way,I have only minor grammatical suggestions:

In the abstract, line 26: Please change the form of the verb.

" to declared the pandemic COVID-19"

The correct form is: to declare the COVID-19 pandemic

Lines 95-98: This sentence is not fluid, I suggest the following: but the presence of such antibodies in the blood serum indicates recent infection with seasonal viruses,  and therefore, the presence of T cell-mediated immunity ensuring partial protection against the severe course of COVID-19.

 Line 124: Please define CHO cells

Line 158: Please define  IPTG

Line 195: Please define PAAG

Line 212: Please define TMB

Line 226: Please define MEM

Line 297: and S5-6 is greater than for (Figure 3A and B, brown lines) than for sera obtained from the 297 Republic of Khakassia (Figure 3A and B , blue lines).

the words than for are repeated. Please correct.

Line 410:  We assumet hat,  please correct.

Line 350: There is evidence that individuals who had had coronavirus infection shortly before the pandemic got sick more easily in the beginning of the COVID-19 pandemic.

The word had is repeated, and I suggest to change "at" instead of "in"

I analyzed your text in Turnitin for similarity and found it has 35%, I suggest paraphrasing the text to reduce the similarity % at least to 25%.

Reviewer 4 Report

Comments and Suggestions for Authors

In this study, Shaprova et al. examined the specificity and virus-neutralizing capacity of antibodies that target the nucleocapsid and spike proteins of SARS-CoV-2 using a collection of sera obtained in October and November 2019, which predates any confirmed cases of COVID-19 in the investigated region. The findings of the study indicate that a portion of the population in the analyzed regions of Russia possessed pre-existing cross-reactive humoral immunity against SARS-CoV-2 before the onset of the COVID-19 pandemic. However, the manuscript should address a few issues.

1. I did not find the results of the virus neutralization assay in the article. Could you provide an explanation for this omission?

2. Please provide information regarding nCoV/Victoria/1/2020, including its genome accession number and pathogenicity.

 3. Please indicate if antibodies were detected in the 18 positive serum samples in the line 113.

 4. Please note that the notation of 'Fig.2' in line 281 is inconsistent with the writing style in line 285. Please review the entire text and make the necessary modifications to maintain consistency.

Round 2

Reviewer 1 Report

Comments and Suggestions for Authors

 I thank the authors for their accurate corrections and additions to the manuscript according to my suggestions. However, I have two more contributions as follows.

1. Table 1: The authors should add the percentage, for example, NI not immune (n;%), 270( %33.79)...

2. Did the authors test the neutralizing activity of SARS-CoV-2 immune sera? They should add this information to the end of the result section.
